# Comparative efficacy and acceptability of interventions for major depression in older persons: protocol for Bayesian network meta-analysis

Tau Ming Liew,[1,2,3] Cia Sin Lee[4]

¹Department of Geriatric Psychiatry, Institute of Mental Health, Singapore
²Psychotherapy Unit, Institute of Mental Health, Singapore
³Saw Swee Hock School of Public Health, National University of Singapore
⁴SingHealth Polyclinics, Singapore

**Correspondence to**
Dr Tau Ming Liew;
tau_ming_liew@imh.com.sg

## ABSTRACT

**Introduction** Major depression is a leading cause of disability and has been associated with adverse effects in older persons. While many pharmacological and non-pharmacological interventions have been shown to be effective to address major depression in older persons, there has not been a meta-analysis that consolidates all the available interventions and compare the relative benefits of these available interventions. In this study, we aim to conduct a systematic review and network meta-analysis to compare the efficacy and acceptability of all the known pharmacological and non-pharmacological interventions for major depression in older persons.

**Methods and analysis** We will search Medline, Embase, PsycINFO, Cumulative Index to Nursing and Allied Health, Cochrane Central Register of Controlled Trials and references of other review articles for articles related to the keywords of 'randomised trial', 'major depression', 'older persons' and 'treatments'. Two reviewers will independently select the eligible articles. For each included article, the two reviewers will independently extract the data and assess the risk of bias using the Cochrane revised tool for risk of bias. Bayesian network meta-analyses will be conducted to pool the depression scores (based on standardised mean difference) and the all-cause discontinuation across all included studies. The ranking probabilities for all interventions will be estimated and the hierarchy of each intervention will be summarised as surface under the cumulative ranking curve (SUCRA). Meta-regression and sub-group analyses will also be performed to evaluate the effect of study-level covariates. The quality of the evidence will be assessed using the Grading of Recommendations Assessment, Development and Evaluation approach.

**Ethics and dissemination** The results will be disseminated through conference presentations and peer-reviewed publications. They will provide the consolidated evidence to inform clinicians on the best choice of intervention to address major depression in older persons.

**PROSPERO registration number** CRD42017075756.

## Strengths and limitations of this study

► This systematic review and meta-analysis will provide a comprehensive summary on the efficacy and acceptability of all available interventions for major depression in older persons.
► The results will provide the highest level of evidence to inform clinicians on the best choice of treatment from among the many available pharmacological and non-pharmacological interventions.
► This protocol has been developed in accordance with the Preferred Reporting Items for Systematic Reviews and Meta-analyses Protocols statement and has been registered with International Prospective Register of Systematic Reviews .
► The overall quality of evidence will be assessed using the Grading of Recommendations Assessment, Development and Evaluation approach.
► This systematic review will be limited to studies which are reported in English language and have been peer reviewed.

medical comorbidities,[3] with reported rates of up to 5% in community-dwelling older persons,[3–5] 5%–10% in primary care[3 6] and as high as 37% after critical care hospitalisations.[3 7] Major depression has a significant impact on the older populations and has been linked to higher risk of suicide,[4] myocardial infarction,[8] stroke,[9] all-cause mortality[4 10] and increasing health services use.[4]

Many of the interventions for major depression in older persons have had recent meta-analyses confirming their efficacy when compared with control groups. These include antidepressants,[11–14] cognitive behavioural therapy,[15] problem-solving therapy,[16] psychological interventions in general[17–19] and the various forms of non-pharmacological interventions.[20–22] However, none of the meta-analyses had compared all the pharmacological and non-pharmacological interventions together to demonstrate the relative benefits of each intervention. It is unknown

## INTRODUCTION
### Rationale

Major depression has been identified by the WHO as one of the leading cause of disability globally.[1 2] In older persons, its prevalence rates rise with the increase in

whether the different types of pharmacological and non-pharmacological interventions have comparable efficacy and are equally suitable for older persons with major depression.

## Objectives

In this study, we aim to conduct a systematic review and network meta-analysis to compare the efficacy and acceptability of all the available pharmacological and non-pharmacological interventions for major depression in older persons. The use of network meta-analysis allows us to pool the evidence on various interventions and rank their benefits relative to each other.[23] It also allows us to conduct indirect comparison of the different interventions, even when there is no direct evidence in the literature to allow head-to-head comparisons.

## METHODS AND ANALYSIS

This protocol is developed in accordance with the Preferred Reporting Items for Systematic Reviews and Meta-analyses (PRISMA) statement.[24 25] It has also been registered with the International Prospective Register of Systematic Reviews (registration number CRD42017075756).

### Eligibility criteria
#### Participants and settings
We will include studies which recruited participants who were:
► 60 years old and above;
► diagnosed with major depression based on formal criteria by the Diagnostic and Statistical Manual of Mental Disorders or International Classification of Diseases;
► having a current episode of major depression (ie, the participants were symptomatic and not in remission at the point of recruitment and the intervention was not intended primarily for the prevention of future relapses).

We will exclude studies which recruited participants with treatment-resistant depression, subthreshold depression, bipolar depression, depression in dementia or psychotic depression. We will not include maintenance studies for major depression as such studies primarily focused on the prevention of relapses in participants who had been asymptomatic or in remission at the point of recruitment.

#### Interventions
We will include studies with pharmacological interventions, including but not limited to[26 27]:
► antidepressants such as citalopram, sertraline, venlafaxine or mirtazapine;
► antipsychotics such as risperidone, quetiapine, olanzapine or aripiprazole;
► mood-stabilisers such as valproate, carbamazepine, lithium or gabapentin.

We will include studies with non-pharmacological interventions, including but not limited to[28–30]:
► psychological interventions such as cognitive behavioural therapy, problem-solving therapy, interpersonal therapy, family interventions or psychodynamic therapy;
► procedural interventions such as electroconvulsive therapy, transcranial magnetic stimulation, transcranial direct-current stimulation or bright light therapy.

We will also include studies which reported on combinations of any of these pharmacological and non-pharmacological interventions.

We will exclude studies which focused primarily on health service models of care but were not related to any modality of intervention, such as studies which evaluated the effectiveness of home treatment, training of general practitioners, multidisciplinary approach or stepped-care approach.

#### Comparators
We will accept control conditions such as placebo intervention, waiting list, treatment as usual, as well as no intervention. We will also include studies with active comparators such as those which compare between two different interventions within the same studies.

#### Outcomes
We will only include a study if it reports the depression scores or the all-cause discontinuation in each study arm following intervention.

#### Study designs and publication types
We will only include randomised controlled trials (RCTs), which aimed to demonstrate the superiority of a treatment to another (also known as superiority trials) and will not include equivalence or non-inferiority trials. The following study designs or publication types will also be excluded: qualitative studies, observational studies, meta-analyses, case reports, case series, ecological studies and policy papers. We intend to include only higher quality evidence and hence will exclude non-randomised trials and publications which were not peer-reviewed (such as conference proceedings, letters and comments).

#### Language and time frame
We will only include studies which are reported in the English language. Apart from that, we do not impose any time restriction to the publication year of the studies. The search of databases will be conducted in January 2018.

#### Information sources and search strategy
We will search Medline, Embase, PsycINFO, Cumulative Index to Nursing and Allied Health and Cochrane Central Register of Controlled Trials for original articles related to the keywords of 'randomised trial', 'major depression', 'older persons' and 'treatments'. Our search strategy for Medline is shown in box. Similar search strategies will be used for the other databases. Additionally, we will also hand search the references of review articles

---

**Box    Search strategy for Medline (via Ovid interface)**

1. *Therapeutics/OR *Drug Therapy/OR *Psychotropic Drugs/ OR *Antidepressive Agents/OR *Antipsychotic Agents/OR *Antimanic Agents/OR *Anticonvulsants/OR *Psychotherapy/ OR *Electroconvulsive Therapy/OR *Transcranial Magnetic Stimulation/OR *Transcranial Direct Current Stimulation/OR *Photatherapy/.

2. (antidepressant* OR "selective serotonin reuptake inhibitor" OR SSRI OR citalopram OR fluoxetine OR paroxetine OR sertraline OR escitalopram OR fluvoxamine OR "serotonin and epinephrine reuptake inhibitor" OR "serotonin epinephrine reuptake inhibitor" OR SNRI OR venlafaxine OR desvenlafaxine OR duloxetine OR milnacipran OR reboxetine OR bupropion OR "noradrenergic and specific serotonergic antidepressant" OR NaSSA OR mirtazapine OR TCA OR tricyclic OR amersergide OR amineptine OR amitriptyline OR amoxapine OR butriptyline OR chlorpoxiten OR clomipramine OR clorimipramine OR demexiptiline OR desipramine OR dibenzipin OR dothiepin OR doxepin OR imipramine OR lofepramine OR melitracen OR metapramine OR nortriptyline OR noxiptiline OR opipramol OR protriptyline OR quinupramine OR trimipramine OR tianeptine OR trazodone OR nefazodone OR agomelatine).ab,ti.

3. (antipsychotic* OR haloperidol OR trifluoperazine OR benperidol OR chlorprothixene OR flupenthixol OR clopenthixol OR chlorpromazine OR prochlorperazine OR sulpiride OR periciazine OR perphenazine OR pimozide OR promazine OR fluspirilene OR methotrimeprazine OR risperidone OR paliperidone OR quetiapine OR olanzapine OR amisulpride OR amisulpiride OR aripiprazole OR clozapine OR sertindole OR zotepine).ab,ti

4. (mood adj stabili*) OR (antimanic adj (agent* OR drug*)) OR anticonvuls* OR anti convuls* OR. carbamazepine OR ethosuximide OR gabapentin OR lacosamide OR lamotrigine OR levetiracetam OR lithium OR oxcarbazepine OR phenobarbital OR phenytoin OR pregabalin OR rufinamide OR tiagabine OR topiramate OR valproic acid OR valproate OR verapamil OR vigabatrin OR zonisamide).ab,ti.

5. (psychotherap* OR therap* OR (cognitive adj behavio* adj therapy) OR "cognitive therapy" OR behavio* adj therapy OR "problem solving therapy" OR "problem-solving therapy" OR "interpersonal therapy" OR "inter-personal therapy" OR (family adj (therapy OR intervention)) OR psychodynamic OR psychoanalytic OR bibliotherapy OR mindful* OR (group adj (therapy OR intervention)) OR emotion-focused OR "emotion focused" OR reminiscen* OR "life review" OR life-review).ab,ti.

6. ("electroconvulsive therapy" OR "electro-convulsive therapy" OR "Transcranial Magnetic Stimulation" OR "Transcranial Direct Current Stimulation" OR "light therapy").ab,ti.

7. #1 OR #2 OR #3 OR #4 OR #5 OR #6.

8. *Depressive Disorder, Major/OR (major adj (depressive OR depression)).ab,ti.

9. *Aged/OR * "Aged, 80 and over"/OR (elder* OR (older adj (person* OR people OR adult*)) OR (late adj life) OR geriatric).ab,ti.

10. *Randomised Controlled Trial/OR (Randomised Controlled Trial).pt OR *Random Allocation/.

11. (Randomised OR randomised OR (random* adj (assigned OR allocated OR assignment OR allocation))).ab,ti.

12. #10 OR #11.

13.  #7 AND #8 AND #9 AND #12.

---

related to the topic to retrieve relevant articles which are not captured through our search of the electronic databases. We will examine the full text of the relevant articles and include the respective articles if they meet our eligibility criteria.

### Study selection

All potential articles will be retrieved and organised in a data management software (Endnote software, Thomson Reuters). After removing duplicate records, two reviewers will independently screen through the titles and abstracts to retain eligible articles. The first 10% of these titles and abstracts will be subjected to a calibration exercise between the two reviewers to ensure mutual agreement.

After completing the screening phase, articles that are deemed as relevant by at least one of the reviewers will be subjected to full-text review. The two reviewers will independently confirm the eligibility of these articles based on the full texts. The first 10% of these full texts will again undergo a calibration exercise by the two reviewers. After the full-text review, the included articles will be used for qualitative synthesis. The chance-corrected agreement between the two reviewers will be assessed using Cohen's Kappa ($\kappa$).

At any point during study selection, the reasons for excluding specific articles will be recorded. Moreover, any disagreements between the two reviewers will be resolved by discussion with a third reviewer.

### Data extraction

Data from the selected studies will be extracted by two reviewers independently and disagreements between the reviewers will be resolved by discussion with a third reviewer. The extracted data will include the following information:

1. Study identification (first author, year of publication, geographic location).
2. Study characteristics (study setting, study design, inclusion criteria, diagnostic criteria of major depression, sample size).
3. Participant characteristics (age, gender, education, number of comorbidities, Mini Mental State Examination score, baseline depression score, depression scale, duration of the current episode of major depression).
4. Characteristics of intervention and comparator (description, treatment dose/intensity, treatment duration, depression score, all-cause discontinuation).

The original authors of the RCTs will be contacted when the required data are not available in the published article.

### Assessment of risk of bias

The risk of bias for each study will be assessed independently by two reviewers using the Cochrane revised tool for Risk of Bias (RoB V.2.0),[31] focusing on biases related to five key domains: randomisation process, deviations from intended interventions, missing outcome

data, measurement of the outcome and selection of the reported result. Each domain will receive a judgement on the risk of bias (high, low or some concerns) and an overall risk of bias will be assigned based on the judgements from the five domains. Any disagreements between the two reviewers will be resolved by discussion with a third reviewer.

## Outcome measures

Our primary outcomes are the efficacy and the acceptability of interventions. The efficacy will be based on the difference in depression scores between the intervention and comparator on the completion of intervention (we will give preference to the primary time point predefined in the original study), computed as standardised mean difference for each RCT. The acceptability will be assessed by the relative risk (RR) of all-cause discontinuation of the intervention. When the information is available, we will also capture a secondary outcome of discontinuation due to adverse effects of interventions and evaluate the RR of discontinuation due to adverse effects. Each intervention will only be grouped by its generic name for pharmacological interventions (such as citalopram, risperidone or valproate) or by its known modality for non-pharmacological interventions (such as cognitive behavioural therapy, problem-solving therapy or electroconvulsive therapy). We will not categorise the interventions further in our analyses of the outcome measures. In the event that the active arm of an RCT involves combinations of interventions, it will be reported as the respective combinations (such as citalopram–cognitive behavioural therapy combination, or risperidone–problem-solving therapy combination).

## Statistical analysis

We will first conduct pairwise meta-analysis provided there are at least two included studies for each pairwise comparison. If there are at least five included studies, we will use the random effects model (DerSimonian and Laird method)[32] to pool the results because this model does not assume homogeneity among the pooled studies. If there are less than five included studies, the random effects model is imprecise in its estimations,[33 34] and we will choose the fixed effect model (Mantel-Haenszel method)[35] instead. We will use the $I^2$ statistic and the Q test to assess heterogeneity in each pairwise meta-analysis. In the presence of substantial heterogeneity ($I^2 > 50\%$)[36] in a particular intervention, we will consider subgrouping the intervention by its dose/intensity and duration and use the subgroups of that intervention in the subsequent network meta-analyses.

We will then conduct the network meta-analyses within a Bayesian framework using the Markov Chains Monte Carlo method. Bayesian analysis provides probabilistic distributions of our estimates of interest through large number of simulations and hence produces results which have more intuitive interpretations. For example, Bayesian analysis generates the 95% credible interval which can be

accurately interpreted as the range containing 95% of the estimates (based on the simulations). In the Bayesian analysis, we will run four Markov chains simultaneously with different arbitrarily chosen initial values and with non-informative priors. Each chain will have at least 10 000 simulations and at least the first 2500 simulations will be discarded as burn-in. Convergence of the simulations will be assessed with the trace plots, kernel density plots and Gelman-Rubin-Brooks plots.

We will employ both fixed-effects and random-effects models in the Bayesian analyses, and will choose the final models based on the deviance information criterion (DIC). While there is no rule of thumb on what constitute significant improvements in DIC, we can take reference from the guideline commonly used in the analogous Akaike Information Criteria[37]: values which are lesser by at least 10 points indicate significantly better model fit and parsimony. Hence, results from the random-effects model will be used if the random-effects model has DIC which is smaller by at least 10 points compared with the fixed-effect model. We will also compare the complexity of model between the fixed-effects and random-effects models using pD (an indicator which has higher value when a model is more complex), with preference for models which are more parsimonious (less complex). The global heterogeneity will be assessed with $I^2$ statistic. A common heterogeneity parameter will be assumed in the random-effects model. Inconsistency between direct and indirect sources of evidence will be statistically assessed using the node-splitting method,[38 39] which generates a P value for the difference between direct and indirect estimates in each closed loop in the network (P values of <0.05 indicates the presence of inconsistency between direct and indirect estimates in a particular closed loop).

We will estimate the ranking probabilities for all interventions and show the results graphically in the form of rankograms and cumulative ranking probability plots. The hierarchy of interventions will be summarised as surface under the cumulative ranking curve (SUCRA) and presented in a scatterplot. SUCRAs have possible values ranging from 0% to 100%, with higher values indicating better efficacy or acceptability. Publication bias will be assessed with comparison-adjusted funnel plot.[40 41]

We will conduct meta-regression analyses to determine whether the results of our network meta-analyses will be affected by the following study-level covariates: sample size, study duration, inclusion criteria, study setting, study design and risk of bias. A covariate is considered as a significant moderator if the 95% credible interval of its beta coefficient in meta-regression does not include the value of zero. If a significant moderator is found, further subgroup analyses will then be conducted to assess the effect of this moderator.

The pairwise meta-analyses will be conducted with STATA V.14. The network meta-analyses will be conducted using JAGS V.4.2.0, through the GeMTC package of R V.3.3.1. The 'Network Graphs' package in STATA statistical software V.14.0 will also used to produce some of

the figures in this study, such as the network plots, ranko-grams, cumulative ranking probability plots and comparison-adjusted funnel plots.[40 42]

## Assessment of quality of evidence

We will use the Grading of Recommendations Assessment, Development and Evaluation approach to report the quality of evidence on efficacy and acceptability of interventions for major depression in older persons. Based on five key domains (methodology quality, directness of evidence, heterogeneity, precision of effect estimates and risk of publication bias), we will classify the quality of evidence in one of four levels—high, moderate, low and very low.[43]

## LIMITATIONS

Several limitations of this study should be noted. First, there can possibly be heterogeneity in the dose/intensity and the duration of each intervention, which may limit the interpretation of the meta-analysis. To address this potential limitation, we will first conduct pairwise meta-analyses to evaluate the amount of heterogeneity using the $I^2$ statistic and the Q test. In the presence of substantial heterogeneity ($I^2 > 50\%$)[36] in a particular intervention, we will consider subgrouping the intervention by its dose/intensity and duration and use the more homogeneous subgroups of that intervention in the subsequent network meta-analyses. In the network meta-analyses, we will also evaluate for inconsistency between direct and indirect estimates using node-splitting method and evaluate for heterogeneity using meta-regression and subgroup analyses. Second, we will exclude non-English and non-peer-reviewed publications (such as conference proceedings and letters). The exclusion of non-peer-reviewed publications is related to our intention of including only higher quality evidence. Regardless, we will monitor the impact of such decision and any possible publication bias using comparison-adjusted funnel plot. Third, we will use all-cause discontinuation as a crude composite measure of treatment acceptability. All-cause discontinuation was chosen (instead of discontinuation due to specific reasons) because this information is more readily available in almost all RCTs, especially among non-pharmacological RCTs where it can be more challenging to clearly attribute the cause of discontinuation to specific reasons such as adverse effects. Hence, the use of all-cause discontinuation will allow us to compare the acceptability of both pharmacological and non-pharmacological interventions within the same model in network meta-analysis.

## ETHICS AND DISSEMINATION

This systematic review will provide the consolidated evidence to inform clinicians on the best choice of intervention, from among the many available options, to address major depression in older persons. This systematic review will be reported in accordance with the recommendations of PRISMA statement for network meta-analyses.[44] It is expected to be completed by January 2020, and the results will be disseminated through conference presentations and publications in peer-reviewed journal.

**Contributors** TML conceived the idea for this systematic review, developed the initial methodology, wrote the first draft and act as the guarantor of the protocol. CSL provided critical feedback on the search strategy, methodology and manuscript. Both authors approved the final version of the manuscript.

**Funding** TML was supported by research grants under the Singapore Ministry of Health's National Medical Research Council (Grant No. NMRC/Fellowship/0030/2016 and NMRC/CSSSP/0014/2017).

**Disclaimer** The funding source had no involvement in any part of the project.

**Competing interests** None declared.

**Patient consent** Obtained.

**Provenance and peer review** Not commissioned; externally peer reviewed.

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
