## [Reviewer comments · BMJ Open]

ARTICLE DETAILS

TITLE (PROVISIONAL)	Comparative efficacy and acceptability of interventions for major depression in older persons: protocol for Bayesian network meta-analysis
AUTHORS	Liew, Tau Ming; Lee, Cia Sin

VERSION 1 – REVIEW

REVIEWER	Fernanda S. Tonin Federal University of Paraná, Curitiba, Brazil
REVIEW RETURNED	11-Oct-2017

GENERAL COMMENTS	This manuscript is a protocol for a Bayesian network meta-analysis. The authors aim to perform a systematic review with network meta-analysis to compare the efficacy and acceptability of all the available pharmacological and non-pharmacological interventions for major depression in older persons. The paper is interesting and scientifically sound; however, I have a couple of major concerns about the methods for conduction the systematic search and the statistical analyses, which the author might want to consider: 1) The searches will be performed in several scientific databases. However, I want to remind you that Embase is 100% covered by Scopus, so please verify the need of using these databases.2) In Box 1 (page 9) – one of the descriptors is repeated: "Depressive Disorder, Major/therapy"[Mesh]. Please check.3) Also in Box 1 (page 9) – why the descriptors for "major depression" and for the population (older persons) will only be searched in article's title? The use of "title/abstract" in the search strategy may enhance the sensibility of your search. Please check if there is no further MeSH Terms that could be added to better describe your population (maybe "aged", "geriatric assessment"). I strongly suggest the revision of the search strategy.4) Please, better describe in method's section how a complementary grey literature search will be performed.5) Why was the outcome of 50% improvement in depression score chosen? This cut-off should be justified.6) Please better define "all-cause attrition" as outcome. Why other outcomes of tolerability (e.g. discontinuation owing adverse events) will not be included in your study?7) Why was a model with non-informative priors for the Bayesian meta-analysis chosen?8) IMPORTANT: I have some concerns about gathering so many different interventions in the same model (e.g. non-pharmacological and pharmacological interventions). How will you guarantee the transitivity in the network meta-analyses? It should be stated in the
--

	manuscript. Also, how do you intend to deal with heterogeneity? Sensitivity analyses will be performed? 9) Will pairwise meta-analyses be conducted before performing the network? 10) Page 13 (statistical analyses) – is the calculation of difference between direct and indirect estimates in closed loop referring to node-split analyses? If so, please state this in the manuscript and better describe its interpretation (e.g. p value) 11) The revised tool for Risk of Bias in randomized trials (RoB 2.0) is already available for use. Consider using this version in your review. 12) The PRISMA-NMA statement and checklists should be cited in your manuscript and should be strictly followed during the conduction and report of your study.
--	---

REVIEWER	Christian Gold Uni Research, Bergen, Norway
REVIEW RETURNED	20-Oct-2017

GENERAL COMMENTS	This manuscript describes the protocol for a systematic review and network meta-analysis of all interventions, pharmacological or non-pharmacological, for major depression in older adults. The authors state that there has not been such a comprehensive review integrating all those interventions. I find the review and statistical methodology appropriate overall. However, there are some concerns that need to be addressed. 1.) The main weakness of the review in its present form lies in the description and categorisation of interventions. At present, many questions remain unanswered, for example:  - Are biological non-pharmacological interventions such as electroconvulsive therapy or deep brain stimulation to be included? - Are community-level interventions to be included? - How are the different types (classes?) of antidepressant medication to be categorised? - The examples of non-pharmacological interventions provided in Rationale and Interventions are slightly different – why? - How are different types of psychosocial interventions to be categorised? - What about different doses of both pharmacological and non-pharmacological interventions? These decisions may have important implications for the results and should therefore be addressed in the protocol, ideally using some existing categorisation (e.g. from published treatment guidelines). 2.) Outcomes (p. 7): How will the reviewers define the “immediate post-intervention” time point for interventions that are (a) single session or single dosage or (b) continued for a long, perhaps indefinite period? 3.) Study design (p. 8): “Conference proceedings” and “letters” are not a design type, but a publication type. Excluding these publication types might contribute to publication bias. 4.) Search strategy (p. 9): I am not sure if this will be sufficient to capture all relevant studies. Search strategies for systematic reviews should be highly sensitive. Has this been checked with an information specialist? The descriptors of population and interventions, but also those of research designs, seem too simple to
---

	me. If no highly sensitive search strategy is used, this should be acknowledged as a limitation. There may also be a small error in #10 (an “OR” that should probably be removed). Overall, the protocol reflects an imbalance between a high level of statistical detail and a low level of clinical detail. The protocol was written by only one author, although the review will be conducted by more than one person. Probably more interdisciplinary collaboration could help to improve the weaknesses of the current protocol.
--	---

VERSION 1 – AUTHOR RESPONSE

RESPONSE TO REVIEWER 1

COMMENT (1):

The paper is interesting and scientifically sound.

RESPONSE:

We thank the reviewer for the encouraging comment.

COMMENT (2):

The searches will be performed in several scientific databases. However, I want to remind you that Embase is 100% covered by Scopus, so please verify the need of using these databases.

RESPONSE:

We thank the reviewer for pointing out the redundancy between the two databases. We have confirmed this ourselves after reviewing the literature on the databases, and have since updated our choice of databases. Between the two databases, we have decided to keep Embase only as we believe that the coverage in Embase should have sufficed for the topic of interest. More importantly, Embase has a much more versatile search engine and can allow the use of Emtree terms (the equivalence of MESH terms) to better refine the search.

COMMENT (3):

In Box 1 (page 9) – one of the descriptors is repeated: "Depressive Disorder, Major/therapy"[Mesh]. Please check.

RESPONSE:

We have corrected the error in our revised search strategy.

COMMENT (4):

Also in Box 1 (page 9) – why the descriptors for “major depression” and for the population (older persons) will only be searched in article’s title? The use of “title/abstract” in the search strategy may enhance the sensibility of your search. Please check if there is no further MeSH Terms that could be added to better describe your population (maybe “aged”, “geriatric assessment”). I strongly suggest the revision of the search strategy.

RESPONSE:

We thank the reviewer for the suggestions to improve on the search strategy. We have taken the advice of the reviewer and completely revised this section. Please see Box 1 in the manuscript for the revised strategy. The key changes are:

- (a) Both MESH terms and keywords are now consistently employed to search for the appropriate population, interventions and research designs, as suggested by the reviewer
- (b) The keyword search now includes both titles and abstracts, as suggested by the reviewer

(c) A more detailed search of the interventions, including the listing of commonly-known interventions

COMMENT (5):

Please, better describe in method's section how a complementary grey literature search will be performed.

RESPONSE:

As suggested, the following has been included in the method's section:

"We will also hand-search the references of review articles related to the topic to retrieve relevant articles which are not captured through our search of the electronic databases. We will examine the full text of the relevant articles and include the respective articles if they meet our eligibility criteria."
(page 9, line 14)

We intend to include only higher-quality evidence which has been peer-reviewed, hence decided not to include "conference proceedings" and "letters". The following sentence has been added to clarify our rationale:

"We intend to include only higher-quality evidence and hence will exclude non-randomized trials and publications which were not peer-reviewed (such as conference proceedings, letters and comments)."
(page 8, line 21)

Notwithstanding, we will test for publication bias in the statistical analysis using comparison-adjusted funnel plot, and evaluate whether publication bias will be a major concern to the findings. We also acknowledged the possible limitation of publication bias in the manuscript:

"... we will exclude non-English and non-peer reviewed publications (such as conference proceedings and letters), which may potentially raise the concern of publication bias. The exclusion of non-peer reviewed publications is related to our intention of including only higher-quality evidence. Regardless, we will monitor the impact of such decision and any possible publication bias using comparison-adjusted funnel plot." (page 16, line 10)

COMMENT (6):

Why was the outcome of 50% improvement in depression score chosen? This cut-off should be justified.

RESPONSE:

We have revisited all our outcome measures and have since removed the above-mentioned as an outcome measure.

COMMENT (7):

Please better define "all-cause attrition" as outcome. Why other outcomes of tolerability (e.g. discontinuation owing adverse events) will not be included in your study?

RESPONSE:

We thank the reviewer for pointing out this ambiguity. As advised, we have revised the measurement of treatment acceptability to all-cause discontinuation of the intervention (page 12, line 20). All-cause discontinuation has been chosen (instead of discontinuation due to specific reasons) because this information is more readily available in most RCTs, especially among non-pharmacological RCTs where it can be more challenging to clearly attribute the cause of discontinuation to specific reasons such as adverse effects. Hence, the use of all-cause discontinuation will allow us to compare the

acceptability of both pharmacological and non-pharmacological interventions within the same model in network meta-analysis. We have also acknowledged this limitation in the manuscript:

“... we will use all-cause discontinuation as a crude composite measure of treatment acceptability. All-cause discontinuation was chosen (instead of discontinuation due to specific reasons) because this information is more readily available in almost all RCTs, especially among non-pharmacological RCTs where it can be more challenging to clearly attribute the cause of discontinuation to specific reasons such as adverse effects. Hence, the use of all-cause discontinuation will allow us to compare the acceptability of both pharmacological and non-pharmacological interventions within the same model in network meta-analysis.” (page 16, line 15).

COMMENT (8):

Why was a model with non-informative priors for the Bayesian meta-analysis chosen?

RESPONSE:

We thank the reviewer for the opportunity to clarify. Non-informative priors do not require us to make assumptions on the prior distributions in Bayesian statistics. Choosing a specific prior distribution can be subjective and may possibly generate misleading results when such information has not been clearly established in the literature.

COMMENT (9):

I have some concerns about gathering so many different interventions in the same model (e.g. non-pharmacological and pharmacological interventions). How will you guarantee the transitivity in the network meta-analyses? It should be stated in the manuscript. Also, how do you intend to deal with heterogeneity? Sensitivity analyses will be performed?

RESPONSE:

We agree with the reviewer that these are pertinent issues that should have been better addressed in the manuscript, and apologize for the oversight. We have since made four key changes as described in the paragraphs below.

(a) We will avoid lumping different interventions together and will only group similar interventions together – by the generic name for each pharmacological intervention (such as mirtazapine, citalopram, quetiapine, valproate or lithium) or by the known modality for each non-pharmacological intervention (such as cognitive behavioural therapy, problem solving therapy, or transcranial magnetic stimulation). We have included the following paragraph to make this point clear to readers:

“Each intervention will only be grouped by its generic name for pharmacological interventions (such as mirtazapine, citalopram, quetiapine, valproate or lithium) or by its known categories for non-pharmacological interventions (such as cognitive behavioural therapy, problem solving therapy, or transcranial magnetic stimulation). We will not categorize the interventions further in our analyses of the outcome measures. In the event that the active arm of a RCT involves combinations of interventions, it will be reported as the respective combinations (such as citalopram–cognitive behavioural therapy combination, or mirtazapine–quetiapine–problem solving therapy combination).” (page 12, line 21)

(b) We will conduct pairwise meta-analysis and evaluate the heterogeneity in each pairwise comparison. In the presence of substantial heterogeneity ($I^2 > 50\%$) in a particular intervention, we will consider sub-grouping the intervention by its dose/intensity and duration, and use the subgroups of that intervention in the subsequent network meta-analyses. We have included the following paragraph to improve clarity to readers:

“We will first conduct pairwise meta-analysis with the random-effects model (DerSimonian and Laird method) provided there are at least two included studies for each pairwise comparison. We will use the I² statistic and the Q test to assess heterogeneity in each pairwise meta-analysis. In the presence of substantial heterogeneity (I²>50%) in a particular intervention, we will consider sub-grouping the intervention by its dose/intensity and duration, and use the subgroups of that intervention in the subsequent network meta-analyses.” (page 13, line 7)

(c) In the network meta-analysis, we will evaluate for inconsistency between the direct and indirect estimates using node-splitting analyses (further explained under COMMENT 10), and evaluate for heterogeneity using meta-regression and subgroup analyses (focusing on the covariates of sample size, study duration, inclusion criteria, study setting, study design and risk of bias).

(d) We have also acknowledged the following limitation in the manuscript:

“Several limitations of this study should be noted. First, there can possibly be heterogeneity in the dose/intensity and the duration of each intervention, which may limit the interpretation of the meta-analysis. To address this potential limitation, we will first conduct pairwise meta-analyses to evaluate the amount of heterogeneity using the I² statistic and the Q test. In the presence of substantial heterogeneity (I²>50%) in a particular intervention, we will consider sub-grouping the intervention by its dose/intensity and duration, and use the more homogeneous subgroups of that intervention in the subsequent network meta-analyses. In the network meta-analyses, we will also evaluate for inconsistency between direct and indirect estimates using node-splitting method, and evaluate for heterogeneity using meta-regression and subgroup analyses.” (page 16, line 1)

COMMENT (10):

Will pairwise meta-analyses be conducted before performing the network?

RESPONSE:

Thank you for the helpful reminder. We have included pairwise meta-analyses in our manuscript, as seen below:

“We will first conduct pairwise meta-analysis with the random-effects model (DerSimonian and Laird method) provided there are at least two included studies for each pairwise comparison. We will use the I² statistic and the Q test to assess heterogeneity in each pairwise meta-analysis.”(page 13, line 7)

COMMENT (11):

page 13 (statistical analyses) – is the calculation of difference between direct and indirect estimates in closed loop referring to node-split analyses? If so, please state this in the manuscript and better describe its interpretation (e.g. p value)

RESPONSE:

Yes, the reviewer has rightly pointed out the method used is “node-splitting analyses”. We have made this clearer in the manuscript:

“Inconsistency between direct and indirect sources of evidence will be statistically assessed using the node-splitting method, which generates a p-value for the difference between direct and indirect

estimates in each closed-loop in the network (p-values of <0.05 indicates the presence of inconsistency between direct and indirect estimates in a particular closed-loop).” (page 14, line 13)

COMMENT (12):

The revised tool for Risk of Bias in randomized trials (RoB 2.0) is already available for use. Consider using this version in your review.

RESPONSE:

Thank you for the suggestion. We have included the revised tool for Risk of Bias (RoB 2.0) in protocol, with the following statements:

“The risk of bias for each study will be assessed independently by two reviewers using the Cochrane revised tool for Risk of Bias (RoB 2.0), focusing on biases related to five key domains: randomization process, deviations from intended interventions, missing outcome data, measurement of the outcome and selection of the reported result. Each domain will receive a judgement on the risk of bias (high, low or some concerns) and an overall risk of bias will be assigned based on the judgements from the five domains.” (page 12, line 7)

COMMENT (13):

The PRISMA-NMA statement and checklists should be cited in your manuscript and should be strictly followed during the conduction and report of your study.

RESPONSE:

Thank you for the helpful pointer. We have added the following statement (together with the relevant reference):

“This systematic review will be reported in accordance with the recommendations of PRISMA statement for network meta-analyses.” (page 17, line 3)

RESPONSE TO REVIEWER 2

COMMENT (1):

I find the review and statistical methodology appropriate overall.

RESPONSE:

We thank the reviewer for the kind comments.

COMMENT (2):

Are biological non-pharmacological interventions such as electroconvulsive therapy or deep brain stimulation to be included?

RESPONSE:

Yes, they will be included if available. To improve the clarity, we have provided examples of some of the possible interventions that we will include:

“We will include studies with non-pharmacological interventions, including but not limited to:

- Psychological interventions such as cognitive behavioural therapy, interpersonal therapy, problem solving therapy, psychodynamic therapy or family interventions;
- Procedural interventions such as electroconvulsive therapy, transcranial magnetic stimulation, transcranial direct-current stimulation or bright light therapy.” (page 7, line 15)

COMMENT (3):

Are community-level interventions to be included?

RESPONSE:

We have no limitations on the studies based on the setting of interventions, as long as they fulfil our inclusion criteria of:

- (1) participants who were 60 years old and above;
- (2) diagnosed with major depression based on formal criteria by the Diagnostic and Statistical Manual of Mental Disorders (DSM) or International Classification of Diseases (ICD); and
- (3) having a current episode of major depression (that is, the participants are currently symptomatic and not in remission; and the intervention was not intended primarily for the prevention of future relapses).

COMMENT (4):

How are the different types (classes?) of antidepressant medication to be categorised?

RESPONSE:

We thank to reviewer for the opportunity to clarify further. The antidepressants will be grouped by their generic names (such as sertraline, escitalopram, mirtazapine or venlafaxine). Network meta-analysis does not require us to further classify the interventions – it allows each intervention to be directly and indirectly compared with the others, as long as they are connected to each other within a network. To improve clarity to readers, we have included the following paragraph:

“Each intervention will only be grouped by its generic name for pharmacological interventions (such as mirtazapine, citalopram, quetiapine, valproate or lithium) or by its known modality for non-pharmacological interventions (such as cognitive behavioural therapy, problem solving therapy, or transcranial magnetic stimulation). We will not categorize the interventions further in our analyses of the outcome measures. In the event that the active arm of a RCT involves combinations of interventions, it will be reported as the respective combinations (such as citalopram–cognitive behavioural therapy combination, or mirtazapine–quetiapine–problem solving therapy combination).” (page 12, line 21)

COMMENT (5):

The examples of non-pharmacological interventions provided in Rationale and Interventions are slightly different – why?

RESPONSE:

The examples in Rationale are those with recent publications on their efficacy, while the examples in Interventions are some of the treatments that have been used to treat major depression. To reduce confusion to the readers, we have revised the Rationale section to the following:

“Many of the interventions for major depression in older persons have had recent meta-analyses confirming their efficacy when compared to control groups. These include antidepressants, cognitive behavioural therapy, problem solving therapy, psychological interventions in general, and the various forms of non-pharmacological interventions.” (page 5, line 13)

We have also revised the Intervention section to the following:

“We will include studies with pharmacological interventions, including but not limited to:

- Antidepressants such as citalopram, sertraline, venlafaxine or mirtazapine;
- Antipsychotics such as risperidone, quetiapine, olanzapine or aripiprazole;
- Mood-stabilizers such as valproate, carbamazepine, lithium or gabapentin.

We will include studies with non-pharmacological interventions, including but not limited to:

- Psychological interventions such as cognitive behavioural therapy, interpersonal therapy, problem solving therapy, psychodynamic therapy or family interventions;
- Procedural interventions such as electroconvulsive therapy, transcranial magnetic stimulation, transcranial direct-current stimulation or bright light therapy.” (page 7, line 10)

COMMENT (6):

How are different types of psychosocial interventions to be categorised?

RESPONSE:

Similar to our response to Comment(4), the different types of psychosocial interventions will only be grouped by their known modalities (for example, cognitive behavioural therapy, problem solving therapy, family interventions, physical exercise, psychodynamic therapy, electroconvulsive therapy, transcranial magnetic stimulation or transcranial direct-current stimulation). Network met-analysis does not require us to further classify the interventions – it allows each intervention to be directly and indirectly compared with the others, as long as they are connected to each other within a network. We have included the following paragraph to improve the clarity to readers:

“Each intervention will only be grouped by its generic name for pharmacological interventions (such as mirtazapine, citalopram, quetiapine, valproate or lithium) or by its known modality for non-pharmacological interventions (such as cognitive behavioural therapy, problem solving therapy, or transcranial magnetic stimulation). We will not categorize the interventions further in our analyses of the outcome measures. In the event that the active arm of a RCT involves combinations of interventions, it will be reported as the respective combinations (such as citalopram–cognitive behavioural therapy combination, or mirtazapine–quetiapine–problem solving therapy combination).” (page 12, line 21)

COMMENT (7):

What about different doses of both pharmacological and non-pharmacological interventions?

These decisions may have important implications for the results and should therefore be addressed in the protocol, ideally using some existing categorisation (e.g. from published treatment guidelines).

RESPONSE:

We agree with the reviewer that this is an important issue that should have been made clear in the protocol, and apologize for the oversight. As mentioned in the previous response, each intervention will be grouped by its generic name for pharmacological interventions or by its known modality for non-pharmacological interventions. However, we will conduct pairwise meta-analysis and evaluate the heterogeneity in each pairwise comparison. In the presence of substantial heterogeneity ($I^2 > 50\%$) in a particular intervention, we will consider sub-grouping the intervention by its dose/intensity and duration, and use the subgroups of that intervention in the subsequent network meta-analyses. We have included the following paragraph to improve clarity to readers:

“We will first conduct pairwise meta-analysis with the random-effects model (DerSimonian and Laird method) provided there are at least two included studies for each pairwise comparison. We will use the I^2 statistic and the Q test to assess heterogeneity in each pairwise meta-analysis. In the presence of substantial heterogeneity ($I^2 > 50\%$) in a particular intervention, we will consider sub-grouping the

intervention by its dose/intensity and duration, and use the subgroups of that intervention in the subsequent network meta-analyses.” (page 13, line 7)

We have also acknowledged this limitation in the manuscript:

“Several limitations of this study should be noted. First, there can possibly be heterogeneity in the dose/intensity and the duration of each intervention, which may limit the interpretation of the meta-analysis. To address this potential limitation, we will first conduct pairwise meta-analyses to evaluate the amount of heterogeneity using the I² statistic and the Q test. In the presence of substantial heterogeneity (I²>50%) in a particular intervention, we will consider sub-grouping the intervention by its dose/intensity and duration, and use the more homogeneous subgroups of that intervention in the subsequent network meta-analyses. In the network meta-analyses, we will also evaluate for inconsistency between direct and indirect estimates using node-splitting method, and evaluate for heterogeneity using meta-regression and subgroup analyses.” (page 16, line 1)

COMMENT (8):

Outcomes (p. 7): How will the reviewers define the “immediate post-intervention” time point for interventions that are (a) single session or single dosage or (b) continued for a long, perhaps indefinite period?

RESPONSE:

We thank the reviewer for pointing out the ambiguity in our outcome measure and have revised to the following:

“The efficacy will be based on the difference in depression scores between the intervention and comparator upon the completion of intervention (we will give preference to the primary timepoint predefined in the original study), computed as standardized mean difference (SMD) for each RCT.”(page , line)

It is possible that there are RCTs that are continued for very long or indefinite period – these studies are invariably maintenance studies in the literature (i.e. the interventions are for maintenance treatment after an acute episode of major depression and are primarily for the prevention of future relapses). Such studies will have been excluded from our systematic review. We strictly only include studies which involve acute treatment of major depression, and has specified a key inclusion criterion of: “having a current episode of major depression (that is, the participants are currently symptomatic and not in remission).” Meta-analysis of maintenance treatment for major depression is also a very pertinent area but will have to be addressed in separate, other study. To improve the clarity of this point to readers, we have amended the eligibility criteria to include:

“... and the intervention was not intended primarily for preventing future relapses....” (page 6, line 23)

“We will not include maintenance studies for major depression as such studies primarily focused on the prevention of relapses in participants who had been asymptomatic or in remission at the point of recruitment.” (page 7, line 4)

COMMENT (9):

Study design (p. 8): “Conference proceedings” and “letters” are not a design type, but a publication type. Excluding these publication types might contribute to publication bias.

RESPONSE:

We thank the reviewer for pointing out the error on publication type and have made the necessary corrections. We intend to include only higher-quality evidence which has been peer-reviewed, hence

decided not to include “conference proceedings” and “letters”. The following sentence has been added to clarify our rationale:

“We intend to include only higher-quality evidence and hence will exclude non-randomized trials and publications which were not peer-reviewed (such as conference proceedings, letters and comments).” (page 8, line 21)

Notwithstanding, we will test for publication bias in the statistical analysis using comparison-adjusted funnel plot, and evaluate whether publication bias will be a major concern to the findings. We also acknowledged this possible limitation of publication bias in the manuscript:

“... we will exclude non-English and non-peer reviewed publications (such as conference proceedings and letters), and may potentially raise the concern of publication bias. The exclusion of non-peer reviewed publications is related to our intention of including only higher-quality evidence. Regardless, we will monitor the impact of such decision and any possible publication bias using comparison-adjusted funnel plot.” (page 16, line 10)

COMMENT (10):

Search strategy (p. 9): I am not sure if this will be sufficient to capture all relevant studies. Search strategies for systematic reviews should be highly sensitive. Has this been checked with an information specialist? The descriptors of population and interventions, but also those of research designs, seem too simple to me. If no highly sensitive search strategy is used, this should be acknowledged as a limitation. There may also be a small error in #10 (an “OR” that should probably be removed).

RESPONSE:

We agree with the reviewer that the search strategies should be highly sensitive and have since taken the effort to completely revamp this section. Please see Box 1 in the manuscript for the revamped strategy. The key changes are:

- (a) The keyword search now includes both titles and abstracts
- (b) A more detailed search of the interventions, including the listing of commonly known interventions
- (c) Both MESH terms and keywords are now consistently employed to search for the appropriate population, interventions and research designs

In designing this latest search strategy, we also had two key considerations related to our eligibility criteria:

(a) The literature abounds with RCTs which recruited participants based on high scores in depression scales alone (commonly termed as “depressive symptoms” or “depression”), without a proper clinical diagnosis of major depression. It is not our interest to capture studies such as these and we needed our search strategy to have at least some specificity in differentiating between these studies and those related to major depression (our eligibility criteria).

(b) It is not uncommon to have controlled trials which are not randomized (especially those conducted in the earlier periods). These non-randomized trials are rather prone to confounding biases (for example, confounding by indication). It is not our interest to capture studies such as these and we needed our search strategy to have at least some specificity in differentiating between non-randomized and randomized trials.

COMMENT (11):

Overall, the protocol reflects an imbalance between a high level of statistical detail and a low level of clinical detail. The protocol was written by only one author, although the review will be conducted by more than one person. Probably more interdisciplinary collaboration could help to improve the weaknesses of the current protocol.

RESPONSE:

We have revised the manuscript to provide more clinical details as suggested by the reviewer. To summarize, we have now included the following key changes:

- (a) more details on the inclusion and exclusion criteria
- (b) examples of pharmacological and non-pharmacological interventions that we will like to include in our systematic review
- (c) a more detailed search strategy, with attempts to include as many examples of interventions as possible within the search strategy
- (d) more elaboration on how the interventions will (or will not) be categorized
- (e) a limitation section to acknowledge some of the potential challenges of this study

As suggested, we have also included a co-author from a different discipline who has provided additional input and perspective to the protocol (page 1, line 4).

VERSION 2 – REVIEW

REVIEWER	Fernanda S. Tonin Department of Pharmacy Federal University of Paraná, Curitiba, Brazil
REVIEW RETURNED	14-Nov-2017

GENERAL COMMENTS	The authors have performed the corrections and considered the reviewer's suggestions for improving the manuscript. Overall, my concerns about the study's goals and statistical analyses were answered. However, I have a couple of minor questions about the reformulated methods: 1) The chosen databases seem now appropriate for the systematic review. Just to clarify, both Scopus and Embase have a broad coverage and both present Emtree terms to better refine the search. To define which database will be used is a criterion of the researchers and takes into account the availability of the database. 2) The search strategy seems more properly designed. I have just some questions: (i) why an asterisks (*) was used before some descriptors? (e.g. *Anticonvulsants); (ii) please check the need for quotation marks for expressions with more than two words (e.g. "selective serotonin reuptake inhibitor"); (iii) please check the need for descriptors of "blinding", because the search can lose sensitivity. 3) Page 16, line 10: please remove the comment "...which may potentially raise the concern of publication bias". Non-English studies are not entire related to low quality or publication bias. 4) Apart from the defined outcome "All-cause discontinuation" it will be important to extract data on specific causes of discontinuation and tolerability (when available in the RCTs). All-cause discontinuation is a broad definition and could lead to misleading results. Moreover, because this outcome is not entire related to
--

	acceptability (e.g. it could be discontinuation owing to inefficacy, loss of criteria or death). Since this study is a protocol, it is possible to over-include some specific outcomes, even if they will not be used in the network meta-analyses. Qualitative syntheses with these data it is also possible. 5) In the network meta-analyses, the RCTs with combination arms (e.g. citalopram–cognitive behavioural therapy combination) will be considered as a combined therapy, i.e. different from non-pharmacological or pharmacological interventions? Please clarify it. How transitivity will be guaranteed with so many different interventions? 6) Why in the pairwise meta-analyses the random-effects model will be used? You can apply both (fixed and random) and then evaluate the results. Which will be the effects measures and statistical methods for assessing the pairwise meta-analyses? Which is the confidence interval used? (I assume that will be 95% CI). Please state this information.
--	--

REVIEWER	Christian Gold Uni Research, Bergen, Norway
REVIEW RETURNED	20-Nov-2017

GENERAL COMMENTS	Overall, the authors have addressed my concerns adequately. I am still not convinced that the boundaries between similar interventions have been thoroughly thought through, but I accept that some of this work may need to be done once all the studies are in. Some minor concerns remain, but these are small enough that I do not need to check the manuscript again. 1.) There are still some inconsistencies in the list of interventions. For example, if I label the psychosocial therapies listed in the introduction as A (cognitive behavioural therapy), B (problem solving therapy), and C (psychological interventions in general), then the list in Methods includes A, D (interpersonal therapy), B, E (psychodynamic therapy), and F (family interventions) – different categories in a different order. The search strategy seems to include some variant of the categories above (with some sub-categories not mentioned elsewhere). In Outcome measures, A and B are listed alone. It is clear that not all categories can be listed in all places, but a consistent list would use the same order throughout and would clarify what are the main classes and which sub-classes belong to which main class. 2.) Some relevant intervention classes such as creative arts therapies (including art therapy, dance movement therapy, and music therapy) may not be captured by the current search strategy. 3.) I still think that reference to existing clinical guidelines would help to clarify what the main intervention classes are (perhaps including the recommended doses). I think that feasibility of the review will be greatly enhanced by having the intervention classes and sub-classes as clear as possible. However, as noted above, I accept that some of this may be part of the process of conducting the review, rather than part of the pre-planning in the protocol.
---

VERSION 2 – AUTHOR RESPONSE

RESPONSE TO REVIEWER 1

COMMENT (1):

The authors have performed the corrections and considered the reviewer's suggestions for improving the manuscript. Overall, my concerns about the study's goals and statistical analyses were answered.

RESPONSE:

We like to take this opportunity to thank the reviewer again for the many useful suggestions which helped to improve the manuscript.

COMMENT (2):

The chosen databases seem now appropriate for the systematic review. Just to clarify, both Scopus and Embase have a broad coverage and both present Emtree terms to better refine the search. To define which database will be used is a criterion of the researchers and takes into account the availability of the database.

RESPONSE:

We thank the reviewer for the kind explanations.

COMMENT (3):

The search strategy seems more properly designed. I have just some questions: (i) why an asterisks (*) was used before some descriptors? (e.g. *Anticonvulsants); (ii) please check the need for quotation marks for expressions with more than two words (e.g. "selective serotonin reuptake inhibitor"); (iii) please check the need for descriptors of "blinding", because the search can lose sensitivity.

RESPONSE:

(i) The asterisk (*) before and the slash (/) after a term (e.g. *Anticonvulsants/) indicates the term is a Mesh term.

(ii) As suggested, we have included quotation marks for expressions with more than two words in the search strategies.

(iii) We thank the review for highlighting this. We have removed search term using "blinding".

COMMENT (4):

Page 16, line 10: please remove the comment "...which may potentially raise the concern of publication bias". Non-English studies are not entire related to low quality or publication bias.

RESPONSE:

We have removed the phrase as suggested.

COMMENT (5):

Apart from the defined outcome "All-cause discontinuation" it will be important to extract data on specific causes of discontinuation and tolerability (when available in the RCTs). All-cause discontinuation is a broad definition and could lead to misleading results. Moreover, because this outcome is not entire related to acceptability (e.g. it could be discontinuation owing inefficacy, loss of criteria or death). Since this study is a protocol, it is possible to over-include some specific outcomes,

even if they will not be used in the network meta-analyses. Qualitative syntheses with these data it is also possible.

RESPONSE:

We thank the reviewer for this useful suggestion, and has incorporated the suggestion as below:

“When the information is available, we will also capture a secondary outcome of discontinuation due to adverse effects of interventions and evaluate the RR of discontinuation due to adverse effects.”
(page 12, line 18)

COMMENT (6):

In the network meta-analyses, the RCTs with combinations arms (e.g. citalopram–cognitive behavioural therapy combination) will be considered as a combined therapy, i.e. different from non-pharmacological or pharmacological interventions? Please clarify it. How transitivity will be guarantee with so many different interventions?

RESPONSE:

RCTs which evaluate combination of pharmacological and non-pharmacological interventions are generally rare in the literature on geriatric depression, as it is more likely for researchers to evaluate one specific intervention at a time. The examples that we gave in the manuscript (e.g. citalopram–cognitive behavioural therapy combination) were primarily to address possible concerns about heterogeneity (and probably more hypothetical in nature), as it may be less appropriate to pool studies on citalopram with studies on citalopram–cognitive behavioural therapy combination. As a hypothetical example, each of the following (if there are available studies) will be an individual node in the network meta-analysis:

Node A: Citalopram

Node B: Cognitive behavioural therapy

Node C: Citalopram–cognitive behavioural therapy

Node D: No intervention

In this case, inconsistency will be assessed if there are both direct and indirect comparisons among these nodes, or acknowledged as a limitation when the required evidence are not available.

COMMENT (7):

Why in the pairwise meta-analyses the random-effects model will be used? You can apply both (fixed and random) and then evaluate the results. Which will the effects measures and statistical methods for assess the pairwise meta-analyses? Which is the confidence interval used? (I assume that will be 95% CI). Please state this information.

RESPONSE:

We have now included plans for both fixed and random effects model in our pairwise meta-analyses, as well as incorporated some of recommendations in the literature on when to use fixed or random effects model (references included). The updated paragraphs read as following:

“If there are at least five included studies, we will use the random effects model (DerSimonian and Laird method)³² to pool the results because this model does not assume homogeneity among the pooled studies. If there are less than five included studies, the random effects model is imprecise in its estimations^{33 34} and we will choose the fixed effect model (Mantel-Haenszel method)³⁵ instead.”
(page 13, line 7)

We described the effects measured under the topic of “Outcome measures” (page 12, line 19) for coherence purposes, as they will be the same for both pairwise and network meta-analyses (we like to apologize if this was not apparent). The conventional practice of 95% CI will be used.

RESPONSE TO REVIEWER 2

COMMENT (1):

Overall, the authors have addressed my concerns adequately. I am still not convinced that the boundaries between similar interventions have been thoroughly thought through, but I accept that some of this work may need to be done once all the studies are in.

RESPONSE:

We thank the reviewer for the kind remarks and understanding.

COMMENT (2):

There are still some inconsistencies in the list of interventions. For example, if I label the psychosocial therapies listed in the introduction as A (cognitive behavioural therapy), B (problem solving therapy), and C (psychological interventions in general), then the list in Methods includes A, D (interpersonal therapy), B, E (psychodynamic therapy), and F (family interventions) – different categories in a different order. The search strategy seems to include some variant of the categories above (with some sub-categories not mentioned elsewhere). In Outcome measures, A and B are listed alone. It is clear that not all categories can be listed in all places, but a consistent list would use the same order throughout and would clarify what are the main classes and which sub-classes belong to which main class.

RESPONSE:

We thank the reviewer for pointing this out and we agree that consistency in our listings will be helpful so as not to confuse readers. We have since made the necessary changes, as far as possible, to keep the list consistent.

COMMENT (3):

Some relevant intervention classes such as creative arts therapies (including art therapy, dance movement therapy, and music therapy) may not be captured by the current search strategy.

RESPONSE:

We acknowledge the limitation and challenges of the current search strategy in including an exhaustive list of therapies for major depression in older persons. We have included the mesh term “Therapeutics” (or “Therapy” in the other databases), which is an encompassing term used in databases to capture all forms of interventions (both pharmacological and non-pharmacological) for diseases.

COMMENT (4):

I still think that reference to existing clinical guidelines would help to clarify what the main intervention classes are (perhaps including the recommended doses).

I think that feasibility of the review will be greatly enhanced by having the intervention classes and sub-classes as clear as possible. However, as noted above, I accept that some of this may be part of the process of conducting the review, rather than part of the pre-planning in the protocol.

RESPONSE:

We appreciate the suggestion by reviewer on referencing to existing clinical guidelines, and agree that this can improve clarity to readers especially when they wish to obtain further information. We

have since made reference to the most recent clinical guideline by the Canadian Network for Mood and Anxiety Treatments (reference 26 to 30).